# Improved Diagnosis of Iron Deficiency Anemia in the Critically Ill via Fluorescence Flowcytometric Hemoglobin Biomarkers

**DOI:** 10.3390/cells12010140

**Published:** 2022-12-29

**Authors:** Mascha Zuther, Marie-Luise Rübsam, Mathias Zimmermann, Alexander Zarbock, Christian Hönemann

**Affiliations:** 1Klinik für Anästhesiologie, Operative Intensivmedizin und Schmerztherapie, Universitätsklinikum Münster, 48149 Münster, Germany; 2Klinik für Anästhesiologie, Intensiv-, Notfall- und Schmerzmedizin, Universitätsmedizin Greifswald, 17475 Greifswald, Germany; 3Institut für Labormedizin Köpenick, DRK Kliniken Berlin, 12559 Berlin, Germany; 4Abteilung für Anästhesiologie und Operative Intensivmedizin, St. Marienhospital Vechta, 49337 Vechta, Germany

**Keywords:** reticulocyte parameters, anemia, iron deficiency, functional iron deficiency, reticulocyte hemoglobin content (RET-H*e*), intensive care

## Abstract

Background: Iron deficiency anemia (IDA) is common in critically ill patients treated in the intensive care unit (ICU), and it can lead to severe consequences. Precise and immediate diagnostics are not available, but they are inevitably needed to administer adequate therapy. Serological parameters such as serum ferritin and transferrin saturation (TSAT) are heavily influenced by simultaneous inflammation reactions, resulting in the need for more suitable parameters. Reticulocyte biomarkers such as reticulocyte hemoglobin content (RET-H*e*) and Delta-hemoglobin equivalent (Delta-H*e*) determined by fluorescence flowcytometry are more specific for the diagnosis of IDA-based anemia and should be investigated for this purpose. Methods: In a prospective cohort single-center study, serum ferritin and transferrin saturation (TSAT) were collected and compared to RET-H*e* and Delta-H*e* by performing a receiver operating curve (ROC) analysis. The sensitivity and specificity of a single variable or the combination of two variables, as well as cutoff values, for the diagnosis of IDA were calculated. A group comparison for IDA patients without IDA was performed for a control group. Results: A total of 314 patients were enrolled from an interdisciplinary ICU. RET-H*e* (area under the curve (AUC) 0.847) and Delta-H*e* (AUC 0.807) did indicate iron-deficient anemia that was more specific and sensitive in comparison to serum ferritin (AUC 0.678) and TSAT (AUC 0.754). The detection of functional iron deficiency (FID) occurred in 28.3% of cases with anemia. Conclusions: Determination of RET-H*e* and Delta-H*e* allows for the increased precision and sensitivity of iron-deficient anemia in the ICU.

## 1. Introduction

Anemia remains a common phenomenon in critically ill patients treated in intensive care, with a mean prevalence up to 65% of all patients at the time of admission and nearly 97% after a length of stay of 8 days [1]. 

The etiology of anemia in critically ill patients is often multicausal, influenced by underlying (chronic) diseases and complicated through relevant blood loss for diagnostic investigations, surgery, or hemolysis [1]. The most common reason for anemia in hospitalized patients is anemia of chronic diseases (ACDs) [2]. ACD arises mainly through dysregulation of iron distribution, with consequent iron-restricted erythropoiesis, triggered by the hepatic hormone hepcidin and the release of proinflammatory cytokines [3]. These frequent inflammatory transmitters lead to functional iron deficiency (FID), characterized by functional hypoferremia through the depressed uptake of nutritional iron and retention of iron into the reticulohistiocytic system (RHS). This is observed even despite heavily filled body iron storages with elevated serum ferritin and hepcidin levels [3,4]. Additionally, a reduced production of erythropoietin, a shortened life span of erythrocytes, and an influenced proliferation of erythroid progenitor cells can enhance the emergence of ACD [3,5].

Both absolute and functional iron deficiency (ID) require a sensitive and timely diagnosis to optimize anemia therapy. The ability of the recommended serological parameter serum ferritin and transferrin saturation in assessing the iron status in the critically ill is frequently limited due to influences of present inflammatory conditions [6,7]. The further detection of surrogate parameters such as C-reactive protein (CRP) is recommended to raise the used cutoff value of serum ferritin to <30 ng/mL or above in such cases [5,8]. Serological parameters such as soluble transferrin receptor (sTfR), its ratio to the logarithm of serum ferritin (ratio sTfR/log ferritin), or hepcidin aim to simplify the diagnosis of IDA, ACD, or combinations of both need standardization, evaluation in prospective study designs, and better laboratory prevalence. Additionally, they are costly [9,10]. 

The quantitative reticulocyte parameters reticulocyte hemoglobin content (RET-H*e*) and Delta-hemoglobin equivalent (Delta-H*e*) directly assess the current iron supply to erythropoiesis independent of body iron storages. RET-H*e* with a reference range from 29.3 to 35.4 pg [11] reflects the mean hemoglobin value (Hb) of the newly formed immature reticulocytes [12,13]. Delta-H*e* is the calculated difference between the mean Hb value of the freshly formed reticulocytes and the mean Hb value of the total amount of mature red blood cells (RBC-H*e*). The values should physiologically be in the positive range, since reticulocytes show a higher Hb content compared to adult erythrocytes [14]. The measured values of RET-H*e* are independent from simultaneous inflammatory conditions, show quick response to changes in iron availability [15], and depict the functional availability of iron for erythropoiesis within a clinically relevant period from 1 to 2 days [16,17]. The additional determination of Delta-hemoglobin equivalent (Delta-H*e*) gives further insight into the dynamics of hemoglobinization of red blood cells over time to analyze real time changes in iron availability and the quality of erythropoiesis, as well as the occurrence of acute inflammation reactions [14,18,19].

We hypothesized that the repeated determination of RET-H*e* and Delta-H*e* enables a more sensitive and accurate detection of iron-deficient anemia types. This would simplify adjusted therapy administration for patients treated in the intensive care unit (ICU), allowing us to detect all types of ID, not just IDA. The study compares RET-H*e* and Delta-H*e* as reticulocyte parameters with diagnostic standards serum ferritin and TSAT for the diagnosis of IDA and FID in intensive care.

## 2. Materials and Methods

### 2.1. Study Design

This prospective observational study was approved by the Ethics Committee of the medical association of Lower Saxony, Germany (Ref: Bo/08/2021, 06.04.2021) and was conducted in accordance with the Declaration of Helsinki.

Over a period of four months in 2021, hematological and serological parameters have been collected from 314 patients aged from 18 to 93 years who stayed in the ICU of Sankt Marienhospital, Vechta, Germany, for a minimum of one day. Prospectively, following informed consent, in-hospital patients from surgical and non-surgical disciplines were enrolled. Exclusion criteria were an age younger than 18 years and incapability of consenting. The Sankt Marienhospital is a regional hospital with 310 beds and a general ICU with 10 intensive and 4 intermediate care beds.

### 2.2. Study Aim

The primary endpoint of the study was to investigate the ability of RET-H*e* and Delta-H*e* to diagnose iron-deficient anemia types in critically ill patients and whether these biomarkers provide a better sensitivity or area under curve (AUC) in receiver operating characteristic (ROC) analysis in diagnosing IDA and FID than the traditionally used serological biomarkers, namely serum iron, serum ferritin, and TSAT.

### 2.3. Definition of Groups and Patient Allocation

Anemia was defined according to the WHO guidelines as an Hb-level lower than 12 g/dL for women and lower than 13 g/dL for men [20].

Serological parameters were used for the discrimination of the iron status. Absolute iron deficiency (ID) was defined by a serum ferritin level <30 ng/mL or serum ferritin level <100 ng/mL if CRP was elevated >5 mg/L. These cutoff values were set for reaching a higher sensitivity and specificity in diagnosing ID and to additionally include phases of functional iron-restricted erythropoiesis in mild inflammatory conditions [21]. The latter are common in patients treated in the ICU, e.g., due to postoperative reactions, infections, or malignant processes [6]. Functional ID (FID) was defined as a combination of TSAT <20% and ferritin >100 ng/mL in patients with a manifest laboratory/chemical or clinical inflammation reaction [3,22]. All patients without ID and anemia were assigned to a control group. An anemia of any other cause than ID was not further specified. 

Depending on the Hb level and iron status, all patients were divided into five study groups for comparative analyses: the control group (no anemia; no ID), latent ID group (no anemia; ID), IDA group (anemia plus ID), FID group (anemia plus FID), and group of others (anemia without ID) (Table 1).

### 2.4. Laboratory Methods

The first patient’s blood sample was collected straight after admission on the ICU and on every following day, as part of the daily routine, for a maximum of five days. The K3 potassium salt of ethylendiamintetraacetic acid (EDTA-K3) S-Monovettes^®^ (SARSTEDT AG & Co.KG, Nümbrecht, Deutschland) with a volume of 2.6 mL was used for the hematological determinations, and S-Monovettes^®^ with lithium heparin and a volume of 7.5 mL were used for the assessment of the serological parameters. All samples were analyzed immediately in the same laboratory. Hematological parameters, including hemoglobin, mean corpuscular volume (MCV), mean corpuscular hemoglobin (MCH), RET-H*e*, and Delta-H*e*, were measured by using fluorescence flowcytometry, using an automated hematological analyzer of the XN-1000 series (Sysmex, Kobe, Japan).

Fluorescence flow cytometry enables the identification and quantification of cell populations based on their fluorescence parameters and light-scattering properties. The blood sample to be analyzed is first diluted to a predefined ratio. The cells are then rendered permeable by use of a detergent and labeled with a specific fluorescence marker that binds to intracellular ribonucleic acids such as those of reticulocytes. As the cells flow past multiple lasers, they are analyzed for their forward and sideward light scatter and side fluorescence light. The analysis of the latter allows for an assessment of the size and granularity of the measured blood cells and, based on the fluorescence light scattering, an identification of the cell population. The forward scatter light correlates directly with the intracellular amount of hemoglobin and can therefore be used to characterize any type of red blood cell population regarding its hemoglobin content [23].

Serum ferritin, serum transferrin, serum iron, CRP, and PCT were measured turbidimetrically on Roche Cobas^®^ pure instruments (Roche Diagnostics, Mannheim, Germany). Serum ferritin and PCT were measured in an immunochemistry analyzer (e402); serum transferrin, serum iron, and CRP were determined in a clinical chemistry analyzer (c303). TSAT was calculated with the following equation:(serum iron (μg/dL)/serum transferrin (mg/dL)) × 70.9(1)

### 2.5. Statistical Analysis

Statistical analyses were performed by using IBM SPSS® Statistics (Version 28.0.1.0, IBM Deutschland GmbH, Ehningen, Deutschland) and MedCalc (Version 20.115, MedCalc Software Ltd., Ostend, Belgium).

All statistical analyses were based on a level of significance of *p* < 0.05 and a confidence interval of 95 percent. Missing figures or figures collected after administration of any kind of anemia therapy were excluded from the analysis.

Data are presented as mean ± standard deviation (SD) or median and interquartile range (IQR) (25%; 75%). The Shapiro–Wilk test was combined with a graphical test of distribution to assess the normality of all continuous variables. All relevant variables were non-normally distributed. Sensitivity and specificity were used to describe the diagnostic capability of the parameters. To verify the diagnostic quality of the reticulocyte parameters for diagnosing IDA or FID, receiver operating characteristic analysis was performed for every subgroup. The method used to assess the predictive power of this test was the area under the curve. Suitable cutoff values for clinical use for RET-H*e* and Delta-H*e* were derived based on the estimated costs of false-positive and -negative and true-positive and -negative decisions and disease prevalence [24]. The false-positive/false-negative cost ratio and the disease prevalence were combined in an equation to calculate a slope (m), and the point where this slope touched the curve in the ROC plot was defined as the optimal cutoff point (optimal criterion). The disease prevalence (*P*) used in the equation was the prevalence of the respective anemia type in the present study population. For our analysis, we reduced the cost fraction to ⅓ based on the incremental cost differences between true and false decisions:(2)$$M=\left(\frac{cost\;of\;false\;positive\;decision-cost\;of\;true\;negative\;decision}{cost\;of\;false\;negative\;decision-cost\;of\;true\;positive\;decision}\right)\times \left(\frac{1-P}{P}\right)$$

Spearman’s correlation was used to evaluate the linear correlations between the different parameters. To compare the diagnostic validity of each parameter per group, the AUC of the different parameters were compared by using the method according to DeLong et al. from 1988 [25]. Kruskal–Wallis’s test was used to compare the central trends of RET-H*e*, Delta-H*e*, serum ferritin, CRP, and PCT between the groups. Post hoc analysis (Dunn–Bonferroni test) assessed the significant differences between the individual groups, and the effect size was described by Pearson’s correlation coefficient (r), using the classification according to Cohen from 1992 [26].

## 3. Results

### 3.1. Patients’ Characteristics

Over the whole period, 670 datasets from 314 patients were collected. At study start, 182 (58%) subjects treated in the ICU already had been anemic according to WHO definitions. From the other 132 subjects who were enrolled, 30 (22.7%) acquired anemia in the ICU or hospital acquired anemia (HAA). Overall, 212 (67.5%) subjects have been discharged from ICU with IDA, FID, or anemia of other cause (25 (11.8%), 60 (28.3%), and 127 (59.9%), respectively). In the group of subjects with anemia, a higher median Apache II Score (14 vs. 9) and number of deaths (14 vs. 2) were registered (Table 2). Table 3 shows the mean distribution of the measured laboratory parameters in the respective subgroups.

### 3.2. Relation between RET-He, Delta-He, and the Parameters of Iron Metabolism

The linear correlations between the reticulocyte, hematological, and serological parameters were compared. RET-H*e* showed a moderate-to-strong positive correlation with the erythrocyte indices MCH and MCV in all groups (Spearman’s correlation coefficient rho (ρ) 0.403–0.736; *p* < 0.01), whereas serum ferritin, serum iron, and TSAT did not. On the other hand, these parameters showed moderate-to-strong positive correlations with the biomarkers of inflammation CRP and PCT (ρ 0.406–0.696, *p* < 0.01. RET-H*e* and Delta-H*e* only moderately correlated with CRP and PCT in a few groups (ρ 0.312–0.430 *p* < 0.01). Linear correlations between the reticulocyte parameters and the serological parameters of the iron metabolism were rare and more common in patients without iron deficiency (ρ 0.199–0.347, *p* < 0.05).

### 3.3. Sensitivity and Specificity of RET-He and Delta-He

The ROC analysis was used to assess the diagnostic performance of RET-H*e* and Delta-H*e* in diagnosing IDA and FID (Figure 1). 

### 3.4. Comparison with Serological Parameters 

The results of the AUC analysis (Figure 2 and Table 4) showed that, for the diagnosis of IDA, serum ferritin and RET-H*e* were the most suitable parameters, without any significant difference between the two parameters. The other parameters were significantly inferior to these two.

The results of the comparison of the AUC for the parameters within the FID group showed the reticulocyte indices to be significantly more suitable in diagnosing FID than the currently widely recommended serological iron parameters, i.e., serum ferritin and TSAT.

### 3.5. Verification of Discrimination between Study Groups

Kruskal–Wallis tests, including post hoc tests, were used to compare the mean RET-H*e*, Delta-H*e*, serum ferritin, and inflammatory markers CRP and PCT between the groups, according to their iron status (Figure 3).

RET-H*e* and Delta-H*e* showed significant differences between the iron-deficient groups (IDA and FID) and every other group. The median values for the IDA and FID groups were, in total, lower than for the non-ID groups. This suggests the diagnostic suitability of both parameters in detecting iron-deficient anemia types and differentiating these anemia types from a non-ID status. Neither reticulocyte parameter could distinguish significantly between IDA and FID.

The serum ferritin values showed significant differences between all groups. The values for the latent ID and IDA group were considerably lower than those of the other groups, whilst the mean serum ferritin value of the FID group was markedly higher compared to every other group. The latter also holds true for CRP and PCT: both showed significantly higher values for the FID group compared to every other group.

## 4. Discussion

Our goal was to verify whether the measurement of RET-H*e* and Delta-H*e* allows for a more sensitive and accurate, but also simpler, detection of iron-restricted erythropoiesis and thus various forms of iron deficiency anemia.

While the rate of anemia may appear to be high in our study, anemia of all causes is very high in ICU settings, climbing to 98% in some studies with a longer ICU stay [27]. The underlying deficiency is often not explicitly diagnosed, let alone a differential diagnosis between absolute and functional iron deficiency performed. In one of the cases where such a diagnosis was performed, it could be shown that, in special ICU units caring mainly for COVID-19 patients, FID can be even more prevalent than in our collective [28].

We could show that both RET-H*e* and Delta-H*e* indeed have considerable use both in absolute iron deficiency and in FID. This was found to be the case even in the presence of simultaneous inflammatory conditions demonstrated by elevated inflammatory parameters in the latter group. Especially RET-H*e* demonstrated a particularly strong specificity for both types of iron deficiency. The calculated optimal cutoff lies within the reference range of 29.3–35.4 pg. It should, however, be noted that reference intervals are derived from healthy populations. In contrast, the cutoff was calculated based on the prevalence of anemia within the study group, as well as the cost of true-negative and -positive and false-negative and -positive decisions. These may well vary between various populations.

The Kruskal–Wallis tests confirmed considerably depressed mean RET-H*e* and Delta-H*e* values for patients with an IDA or FID compared to patients without an ID. Additionally, the Spearman correlation suggests that both parameters are strong predictors of IDA that may not have manifested yet. In contrast, since the serological parameters do not really reflect the iron supply for erythropoiesis and are confounded by inflammation, they did not show any correlation with erythrocyte indices. The comparison of the AUCs showed the superiority of RET-H*e* and Delta-H*e* for detecting FID to all serological parameters and the noninferiority of RET-H*e* to serum ferritin for IDA.

The current gold standard for evaluating the iron status of the body is bone-marrow aspiration and staining in Perl’s Prussian Blue [29]. In the context of standard diagnostics, however, this method is irrelevant due to being highly invasive and painful. It also comes with a high turnaround time, cost, and personnel expenditure. The WHO consequently currently suggests serum ferritin as a marker, with a suggested cutoff of <15 ng/mL as the limit for iron-restricted erythropoiesis [30]. Because of its function as an acute phase protein, serum ferritin is heavily biased by simultaneous inflammation reactions and can reach values up to 3000 times of its normal range [7]. To address this challenge, the measurement of surrogate parameters for inflammatory reactions such as CRP is suggested, but the data regarding recommendations for adjusted cutoff points for serum ferritin are ambiguous, reaching from cutoff recommendations up to <100 ng/mL to a combination with the measurement of TSAT, with suggested cutoff points between <16% and <20% [5,31,32,33]. This is particularly problematic in the context of the critically ill since inflammatory reactions are quite common in these patients. Despite that fact, there is, as of now, no clear recommendation for a specific diagnostic procedure in critically ill patients in intensive care.

At the same time, once manifest, anemia exacerbates the outcome of other associated disorders [4,21] and reduces the individual’s quality of life. In addition, it leads to worse patient outcomes and increased costs to the healthcare system. The administration of packed red blood cells to treat anemia, on the other hand, comes with risks of its own [34,35]. To both limit these risks and use the limited supply of packed red blood cells more responsibly, patient blood management was developed. Its goal is to ensure earlier and better diagnostics of anemia and more specific therapy tailored to the individual patient [36]. In our context, this means that anemia associated with iron deficiency should be treated first-line with iron replacement therapy [37]. The two new parameters RET-H*e* and Delta-H*e* can be measured within the determination of a complete blood count with reticulocyte count [38]. This saves patient blood and indicates the need for iron substitution at an early stage, avoiding the manifestation of full anemia and the associated need for packed red blood cells [39,40,41].

Our study could confirm the suitability of RET-He as a reliable parameter even under the aggravated diagnostic circumstances of the ICU. The additional determination of Delta-H*e* further increases the diagnostic sensitivity and the ability to detect early stages of iron deficiency. It also helps in making the interpretation independent of interindividual variations in RET-H*e* by providing a measurement independent of the absolute value of that one parameter. The HemaPlot proposed by Weimann et al. can be used as graphical aid for the joint interpretation of RET-H*e* and Delta-H*e* values, because it directly assigns the measured iron supply of erythropoiesis to diagnostic categories such as ACD [14]. 

While our study shows that ferritin has a comparable AUC for the detection of IDA, this should be seen in the context of our use of ferritin as a marker for group allocation. This was performed based on the WHO recommendations mentioned above [30], and in terms of assessing iron supply to hematopoiesis, could only have been avoided by resorting to bone marrow sampling. The use of this method was dismissed based on its requiring a highly invasive procedure that would not have been performed on the patient anyway. In addition, it has been shown previously that RET-H*e* also correlates with iron in the bone marrow [42]. In general, this issue, while statistically problematic, is difficult to avoid when assessing diagnostic parameters measuring physiologically related phenomena. Therefore, however, it is likely that ferritin performance was overestimated in this study. On the other hand, if that was the case, it would only mean that Ret-H*e* is not just non-inferior but superior to ferritin in diagnosing iron deficiency. It may, however, also mean that some degree of bias was introduced into the calculation of optimal cutoffs. There might be a use for ferritin in the distinction of IDA from FID in a secondary differential diagnosis, in which the reticulocyte parameter failed in our study, even though it has already been described in other study publications [14].

There are further limitations to this study: Our data collection was restricted to a monocentric survey; the results thus require further external validation by multicentric studies. The probability of an introduction of significant bias was reduced by the prospective data collection, defined blood-sample-collection standards, and immediate sample evaluation in the same laboratory. 

A variety of studies and guidelines already exist regarding the optimal iron status determination, but studies investigating these parameters in the intensive-care setting are rare. For example, a recent meta-study which suggested the superiority of TSAT explicitly excluded patients with chronic disease characteristic for the critically ill [12]. This study aims to focus on this setting and to design a guideline for iron-deficient anemia diagnosis in critically ill patients. RET-H*e* and Delta-H*e* were quite suitable parameters for a first-step, routine diagnostic in anemic patients in the ICU. Both parameters can be determined simultaneously with a lowered Hb and in parallel to a marker for inflammatory reactions, e.g., CRP. This allows for an immediate check of the most common anemia causes [43,44]. As already suggested for routine hematology, if more complicated anemia compositions are present, the determination of the serological parameter serum ferritin and TSAT should be considered as a second step [45]. 

## 5. Conclusions

RET-H*e* and Delta-H*e* both showed promising diagnostic capabilities to diagnose IDA and FID in critically ill patients in the ICU. The integration of both parameters into the standard diagnostic of iron deficient anemia is reasonable and enables the detection of different iron deficient anemia types, even with simultaneous inflammation conditions.

## Figures and Tables

**Figure 1 cells-12-00140-f001:**
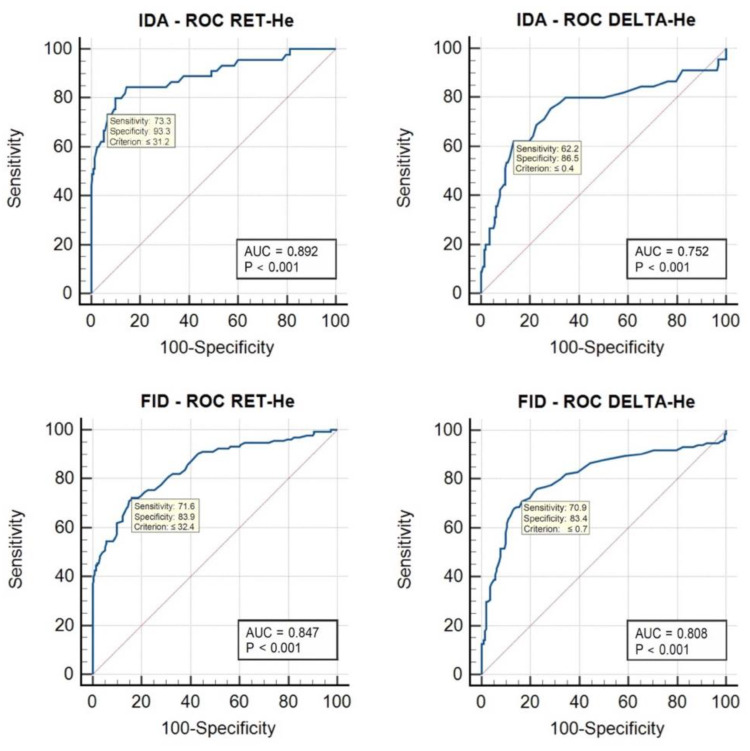
Illustrated are the receiver operating characteristics (ROCs) of RET-H*e* and Delta-H*e* for the group of patients with IDA and FID. The calculated optimal cutoff points (criterion) with their sensitivity and specificity, the area under the curve (AUC), and the level of significance (P) are also included.

**Figure 2 cells-12-00140-f002:**
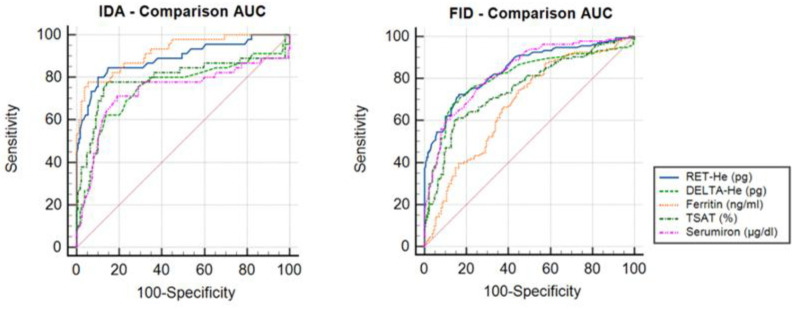
Comparison of the AUC in the IDA group and FID group.

**Figure 3 cells-12-00140-f003:**
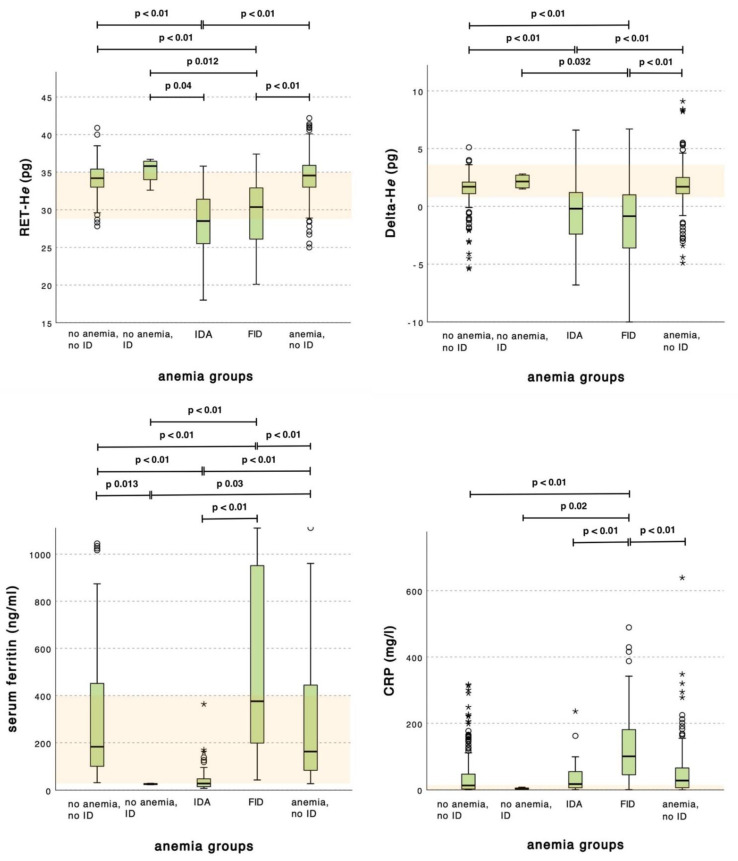
Ability of RET-H*e*, Delta-H*e* serum ferritin, and CRP to discriminate between anemic, iron-deficient, and inflammatory states. Group comparison by Kruskal–Wallis. The colored band marks the reference range; ○, mild outliers (value between 1.5 to 3 IQR from the box); ∗, extreme outliers (value more than 3 IQR from the box); p, *p*-value.

**Table 1 cells-12-00140-t001:** Criteria for patient allocation to study groups.

Study Group	Definition	Function
Patients without ID or anemia	♂: Hb > 13 g/dL ♀: Hb > 12 g/dL Ferritin > 30 ng/mL	control group
Patients with ID without anemia	♂: Hb > 13 g/dL ♀: Hb > 12 g/dL Ferritin < 30 ng/mL	latent ID
Patients with IDA *(Including patients with* *inflammation)*	♂: Hb < 13 g/dL ♀: Hb < 12 g/dL Ferritin > 30 ng/mL OR *Ferritin < 100 ng/mL or* *TSAT < 20%* *if CRP > 5 mg/L and* *Ferritin 100–300 ng/mL*	absolute iron deficiency anemia
Patients with FID and anemia	♂: Hb < 13 g/dL ♀: Hb < 12 g/dL Ferritin > 100 ng/mL AND TSAT < 20% ± CRP > 5 mg/L ± clinical diagnosis	functional iron deficiency anemia
Patients with anemia without ID	♂: Hb > 13 g/dL ♀: Hb > 12 g/dL TSAT > 20%	no iron deficiency

This Table summarizes the criteria for the patient allocation to the study groups. CRP, C-reactive protein, FID, functional iron deficiency; Hb, hemoglobin; ID, iron deficiency; IDA, iron-deficiency anemia; TSAT, transferrin saturation; g, gram; dL, deciliter; mg, milligram; ng, nanogram.

**Table 2 cells-12-00140-t002:** Demographic data of the study population.

Characteristics	*n* = 314	Variance
Age (years) *	64 (19–93)	IQR 52; 75
Female *n* (%)	133 (42.4)	
Male *n* (%)	181 (57.6)	
Body mass index (BMI) (kg/m^2^) #	27.7 (16–48)	± SD 5.6
Length of intensive care unit (ICU) stay (d) *	2 (1–29)	IQR 1; 5
Apache II Score (score value) * Patients without anemia Patients with anemia	13 (0–40) 9 (0–30) 14 (0–40)	IQR 8; 19 IQR 5; 12 IQR 9; 20
Death *n* (%) Patients without anemia Patients with anemia	16 (5) 2 (12.5) 14 (87.5)	
Patients with anemia Patients with anemia at time of admission Patients with hospital acquired anemia (HAA)	212 (67.5) 182 (58) 30 (9.6)	
Patients with proven sepsis	23 (7.3)	
Underlying disease at admission *n* (%) surgical cardiovascular respiratory Patients with COVID-19–Pneumonia others (gastrointestinal, neurological, psychiatric, gynecological, geriatric)	115 (36.6) 94 (29.9) 45 (14.3) 27 (8.6) 60 (19.1)	

This Table shows the demographic data in the respective groups, as well as their variance. Parameters marked with * are parametrically distributed and described with the median (total range) and IQR: 25% and 75%. Parameters marked with # are normally distributed, and their mean (total range) and ±SD are given. Apache II Score is an ICU mortality prediction score, ranging from minimum 0 to maximum 72 points. BMI, body mass index; ICU, intensive care unit; HAA, hospital acquired anemia; kg, kilogram; m, meter.

**Table 3 cells-12-00140-t003:** Laboratory data of the study population.

	Controls *n* (%) 128 (37,2)	Latent ID *n* (%) 4 (1,2)	IDA *n* (%) 25 (7,3)	FID *n* (%) 60 (17,4)	Others *n* (%) 127 (36,9)
Hb (g/dL) *	13.7 (13.1; 14.4)	12.4 (12.1; 12.5)	9.5 (8; 11.4)	9.7 (8.7; 10.8)	10.6 (8.8; 11.7)
MCV (fL) *	88.5 (85.9; 91.7)	89.2 (85.5; 93.5)	83.3 (79; 86.6)	87.1 (82.4; 91.2)	90.6 (86.8; 93.9)
MCH (pg) *	30.1 (28.9; 31.2)	30.4 (28.7; 32)	27.2 (24.8; 29)	28.9 (27.2; 30.4)	30.6 (29.4; 31.7)
RET-H*e* (pg) * RET-H*e* (pg) #	34.2 (33; 35.4) 34.2 (±2.0)	35.8 (33.3; 36.6) 35.2 (±1.8)	28.5 (25.5; 31.6) 28.1 (±4.6)	30.4 (26.1; 33) 29.5 (±4)	34.6 (33; 35.9) 34.3 (±2.8)
Delta-H*e* (pg) * Delta-H*e* (pg) #	1.7 (1.1; 2.1) 1.4 (±1.5)	2.2 (1.5; 2.8) 2.2 (±0.6)	−0.2 (−2.5; 1.3) −0.6 (±3.5)	−0.9 (−3.6; 1) −1.2 (±3.2)	1.7 (1.1; 2.5) 1.8 (±2)
Serum ferritin (ng/mL) * Serum ferritin (ng/mL) #	183.2 (100.6; 452.1) 524.1 (±1041.3)	25.9 (23.2; 28.5) 25.9 (±2.7)	28.4 (14.8; 51.5) 50.8 (±63.2)	375.9 (196.1; 951.8) 958.3 (±1547.5)	163.4 (82.8; 448.7) 620.1 (±2014.8)
TSAT (%) *	22 (14.7; 32.1)	20.4 (11.8; 29.2)	8.5 (5.4; 11.9)	11.5 (7.7; 18.9)	22.6 (15.5; 35.8)
Serum iron (μg/dL) *	61.5 (39.8; 89)	73.5 (47.3; 87.8)	27 (17.5; 43.2)	22 (14; 41)	50 (34; 84.6)
CRP (mg/L) *	13.4 (2.6; 47.8)	3 (2.1; 6.9)	17.7 (6; 57.9)	100.9 (44.4; 182.9)	28.1 (6.5; 66.8)
PCT (ng/dL) *	0.1 (0.0; 0.2)	0.1 (0.0; 0.2)	0.1 (0.0; 0.3)	0.6 (0.2; 2.6)	0.1 (0.0; 0.5)
Leukocytes (10^3^/μL) #	11.5 (±4.2)	12.2 (±3.1)	10.3 (3.3)	13.9 (±6.8)	11.1 (±6.4)

This Table shows the mean distribution of the laboratory parameters in the respective groups, as well as their variance. Except for the leukocytes, all parameters are parametrically distributed and described with the median (IQR: 25% and 75%) *. For the fields marked with #, the mean (±SD) is given. The overall number of patients is higher than the numbers of patients included in the study because of the patients who acquired an HAA and changed groups. ID, iron deficiency; IDA, iron deficiency anemia; FID, functional iron deficiency; Hb, hemoglobin; MCV, mean corpuscular volume; MCH, mean corpuscular hemoglobin; RET-H*e*, reticulocyte hemoglobin equivalent; Delta-H*e*, Delta-hemoglobin equivalent; TSAT, transferrin saturation; CRP, C-reactive protein; PCT, procalcitonin; g, gram; dL, deciliter; μg, microgram; mg, milligram; ng, nanogram; pg, picogram; fL, femtoliter.

**Table 4 cells-12-00140-t004:** Data of the area under the curve (AUC) comparison for the IDA and FID group.

	AUC (IDA):		AUC (FID):
Ferritin RET-H*e* TSAT Delta-H*e* Serum Iron	0.920 0.892 0.791 0.752 0.737	RET-He Serum iron Delta-H*e* TSAT Ferritin	0.847 0.835 0.808 0.754 0.678
**Significant differences:**		**Significant differences:**	
RET-H*e*/Delta-H*e* RET-H*e*/TSAT RET-H*e*/Serum Iron Delta-H*e*/Ferritin Ferritin/TSAT Ferritin/Serum Iron TSAT/Serum Iron	*p* < 0.001 *p* = 0.046 *p* = 0.01 *p* = 0.002 *p* = 0.018 *p* = 0.001 *p* < 0.001	RET-H*e*/Ferritin RET-H*e*/TSAT Delta-He/Ferritin Ferritin/Serum iron TSAT/Serum iron	*p* < 0.001 *p* = 0.009 *p* < 0.001 *p* < 0.001 *p* < 0.001

This Table lists the AUC by decreasing size for the IDA and FID groups. All significant differences between the AUC of different parameters are presented with their *p*-value. AUC, area under the curve.

## Data Availability

The data presented in this study are available upon request from the corresponding author. The data are not publicly available due to privacy restrictions.

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
