# Peer review of "Improved Diagnosis of Iron Deficiency Anemia in the Critically Ill via Fluorescence Flowcytometric Hemoglobin Biomarkers"

_cells, 2022, doi:10.3390/cells12010140_

Round 1

Reviewer 1 Report

The manuscript entitled “Fluorescence flowcytometry reticulocyte biomarkers enable rapid monitoring of iron deficient anemia and infection in intensive care unit patients” reports the sensitivity and specifity of alternate diagnostic for iron deficiency anemia. The FACS determination of the hemoglobin content of reticulocytes is a rapid, inexpensive method insensitive to changes from inflammatory signals. The suitability of this method is new and tested adequately.

However, the manuscrit needs a major revison, a language check by a native speaker, restructuring and shortening a great deal.

Major criticism

The discrimination of various anemia forms and the monitoring of the anemia therapy in the ICU is a topic that urgently should be addressed. The authors of this article use the hemoglobin content of reticulocytes and the difference to the mean hemoglobin concentration of red cells measured by flowcytometry.

The manuscript entitled “Fluorescence flowcytometry reticulocyte biomarkers enable
rapid monitoring of iron deficient anemia and infection in intensive care unit patients” reports the sensitivity and specifity of an alternate diagnostic method for iron deficiency anemia. The FACS determination of the hemoglobin content of reticulocytes is a rapid, inexpensive method insensitive to changes from inflammatory signals. The suitability of this method is new and tested adequately.

The design is a prospective observational cohort study on a single center interdisciplinary ICU for the comparison of the sensitivity and specificity of diagnostic variables Ret-He and Delta-He in comparison to standard-of-care serum iron, ferritin, and TSAT.           

However, the condition of the manuscript does not allow its publication as it is.  The manuscript needs for better readability and adjustment to the scientific level of the journal a thorough major revision, a language check by a native speaker including style and punctuation. Redundant information and superfluous phrases are tiring and should be rigorously erased. It should be condensed to a half of its actual volume. The hypothesis as well as the endpoints of the study, the study form are not clearly expressed but need to.

Minor criticism

Scientific writing is not easy. Precise wording without manipulation or hidden interpretation is obligatory. Furthermore, the manuscript should be given to a native speaker for language check. Do not use so many relative clause ..., which lead to an iron… or which causes functional … which simplifies…. Better to start a new but shorter sentence.

The title -

-is a bit bulky. Try to concentrate, for example “intensive care unit patients” could be “critical ill”. I suggest: “Improved diagnosis of iron deficiency anemia in critical ills by flowcytometric monitoring of reticulocyte’s hemoglobin content.”

Throughout the ms: Use common terms such as iron deficiency anemia (IDA) or anemia of chronic disease (ACD).  Is there a difference between ACD as opposed to anemia of infection or acute inflammation? If so, give a definition with a reference. To my knowledge, anemias are categorized different by pathogenesis, red cell morphology, and clinical presentation. For diagnostic purposes, it would be helpful to lean on the gastroenterological classification such as in  https://www.ncbi.nlm.nih.gov/pmc/articles/PMC2754510/.

The term “functional iron deficiency FID” needs a reference too.

Abstract:

Needs to be condensed, see suggestion:

Background: Iron deficiency anemia (IDA) is common in critically ill patients treated 13 in the intensive care unit (ICU), yet leading to severe consequences. Precise and immediate diagnostics are not available. inevitable to administer adequate therapy. Serological parameters like serum ferritin and 15 transferrin saturation (TSAT) are heavily influenced by simultaneous inflammation reactions, resulting in the need for more suitable parameters. Reticulocyte biomarkers like hemoglobin content (RET-He) and Delta-hemoglobin equivalent (Delta-He) determined by fluorescence flowcytometry are more specific for the diagnosis of IDA-based retic- 18 ulocyte biomarkers from blood count should be investigated for this purpose. Methods: In a prospective cohort single center study, serum ferritin and transferrin saturation (TSAT) were collected and compared to RET-He and DELTA-He by ROC-analysis. Sensitivity and specifity of a single variable or the combination of 2 variables as well as cutoff-values for the diagnosis of IDA were calculated. Group comparisons for IDAPatients without IDA were assigned to a control group. Results: Enrolled were 314 patients from an interdisciplinary ICU. RET-He (Area under curve (AUC) 0.846) and Delta-He (AUC 0.807) did indicate IDA more specific () and sensitive () in comparison to serum ferritin (AUC 0.678) and TSAT (AUC 0.754). The detection of functional iron deficiency (FID) occurred in nnn % of cases without anemia. Conclusions: Determination of RET-He and Delta-He allows an increased precision and sensitivity of IDA in the ICU.  

Introduction section:

The section is lengthy although the diagnostic problem is well described. It needs to be condensed to its half.

Line 32: Avoid unscientific wording such as “Even today”. It is not obvious what do you mean by this or to which condition this expression is directed. It is ok to start the Sentence with “Anemia…”

Line 99-102: This statement is your hypothesis, right? So name it: We hypothesize that…

Line 103-106: Then erase the phrase “To facilitate….”, continue “This study compares …to…

Materials and Methods:

This section should be reorganized. Details on the FACS methodology should be described here. Furthermore, the paragraphs and titles need to be clear and comprehensive. For example, you are giving the group allocation within the statistical analysis paragraph, you are mentioning a control group in the classification of anemia.  I suggest to arrange paragraphs for study design and study aim, definition of groups and allocation, lab methods, statistical analysis.

Line 149- either you define “diagnostic performance”, or you be more precise especially for the studies endpoint. The reader wants to know exactly what the primary endpoint of your study is.

Line 126-131: You devide the subjects of your cohort in diagnostic groups- by which criteria? What is the background to differentiate between ID with out anemia and FID? The group allocation in lines 126-131 came without a reference. Are you trying to tell nutritional from inflammatory based ID? What is the intention behind this separation? It is possible to tell by a combination of Ferritin and TSAT from regular ID without anemia. Explain to the reader why this FID is important to identify and to differentiate from the regular ID without anemia.

Lines 157ff : By which statistical procedure the group comparison was done? Multiple group comparison by Kruskall-Wallis should be mentioned here already.

Line 173-4- erase the not correct formatted citation in paracentesis. Zweig and Campell are citation 23. Explain in brief how this method works.

Results:

This paragraph is in general too long. You don’t have to mention the statistical method before you present the results (i.E. the first sentence of 3.2. “Spearman….).

Avoid redundancy. All interpretation and data presented as graphs should be erased.

The results have to be presented objectively and neutral without explanation or interpretation “ because of their function as acute phase proteins”)- you can comment in the discussion section.

Avoid the word “significant”. You defined a level of significance before and that is it.

First paragraph: Again, avoid redundancy: Erase length text passages with a lot of numbers that better are given in a table. The underlying illness such as septic or cardiogenic shock or the leading ICD of your heterogenic collective should be mentioned.

Table 1 appears to be all mixed up since alignment and centration errors affect the readability.

Use table 1 for the disposition of demographic data, add BMI, Apache-Score or your score for illness severity in Germany, length of ICU stay at enrolment, for example- everything with a connection to the development of anemia.

Use a second table for lab parameters only.

Rename the paragraphs from 3.3 to 3.5 (the use of paragraph titles is not common) by their clinical meaning instead by the method. 3.3. deals with sensitivity and specificity of Ret-He and Delta-He. Discard all text that duplicates the information of the ROC-curve graphs- for example lines 232-239. Erase the part “…, even with simultaneously ongoing inflammatory reactions” in lines 240-241, since you don’t show correlations to inflammation variables in those graphs. There are double dots in the legends graph at the end line 245.

Figure 1 Explain that the term criterion

In 3.4 you compare the sensitivity in relation to the specificity of the various variables for the differential diagnosis of IDA and FID. Please shorten the text from line 247 to 268 and erase non-neutral comments. Be describing and explaining instead of advertising. Shift comments such as “For methodical reasons, we must assume that the diagnostic power of serum ferritin is overestimated in this study design, because of the use of this parameter in stratification for the group assignment.“ (lines 253-258) to the discussion. Include the data information in the graphs and legend of Fig 2.

In 3.5 you try to demonstrate the superiority of Ret-He and Delta-He for the differentiation of various combinations of anemia and (functional) iron deficiency and their independency of inflammation. For the reader, this paragraph needs more simplification, less data included in the text and concentation, less interpretation. Shift everything

Legend Fig 3. Reword in something like this: Ability of Ret-He , Delta-He, ferritin and CRP to identify anemic and iron deficient states. Group comparison by Kruskall-Wallis. The colored band marks the reference range.

Discussion:

In general, you want to say that in comparison of Ferritin and TSAT,  RET-He and Delta-He were able to identify iron deficiency and IDA independent from inflammation. Please rearrange the whole discussion, erase the paragraph titles. First give a short summary of your findings, then discuss why this is important, what is congruent with your findings, is there literature not supporting your results. Then discuss in detail the potential methodological flaws of your method- is it ok to select of group allocation based on parameters that are giving a correlation with the new parameters?

This section has to be convincing and a pleasure to read!

Author Response

Reviewer 1:

We thank the reviewer for their valuable suggestions how to improve the article. Regarding the individual points, we have addressed them as follows.

Major criticism

However, the condition of the manuscript does not allow its publication as it is.  The manuscript needs for better readability and adjustment to the scientific level of the journal a thorough major revision, a language check by a native speaker including style and punctuation. Redundant information and superfluous phrases are tiring and should be rigorously erased. It should be condensed to a half of its actual volume. The hypothesis as well as the endpoints of the study, the study form are not clearly expressed but need to.

We have taken the specific points mentioned below to heart and have addressed them as follow:

Minor criticism

Scientific writing is not easy. Precise wording without manipulation or hidden interpretation is obligatory. Furthermore, the manuscript should be given to a native speaker for language check. Do not use so many relative clause ..., which lead to an iron… or which causes functional … which simplifies…. Better to start a new but shorter sentence.

 Some of the relative clauses were resolved through rephrasing or eliminating the respective passage entirely, others were resolved individually.

The title -

  • -is a bit bulky. Try to concentrate, for example “intensive care unit patients” could be “critical ill”. I suggest: “Improved diagnosis of iron deficiency anemia in critical ills by flowcytometric monitoring of reticulocyte’s hemoglobin content.”

We rephrased the title in a similar fashion

  • Throughout the ms: Use common terms such as iron deficiency anemia (IDA) or anemia of chronic disease (ACD).  Is there a difference between ACD as opposed to anemia of infection or acute inflammation? If so, give a definition with a reference. To my knowledge, anemias are categorized different by pathogenesis, red cell morphology, and clinical presentation. For diagnostic purposes, it would be helpful to lean on the gastroenterological classification such as in  https://www.ncbi.nlm.nih.gov/pmc/articles/PMC2754510/.

The gastroenterological classification cited focuses more on aspects not related to iron status, whereas we specified we focus on the differentiation between iron deficient and non-iron deficient anemia, not further characterizing the latter. Nonetheless, we have sought to clarify the issue.

  • The term “functional iron deficiency FID” needs a reference

 Done

Abstract:

Needs to be condensed, see suggestion:

  • Background: Iron deficiency anemia (IDA) is common in critically ill patients treated 13 in the intensive care unit (ICU), yet leading to severe consequences. Precise and immediate diagnostics are not available. inevitable to administer adequate therapy. Serological parameters like serum ferritin and 15 transferrin saturation (TSAT) are heavily influenced by simultaneous inflammation reactions, resulting in the need for more suitable parameters. Reticulocyte biomarkers like hemoglobin content (RET-He) and Delta-hemoglobin equivalent (Delta-He) determined by fluorescence flowcytometry are more specific for the diagnosis of IDA-based retic- 18 ulocyte biomarkers from blood count should be investigated for this purpose. Methods: In a prospective cohort single center study, serum ferritin and transferrin saturation (TSAT) were collected and compared to RET-He and DELTA-He by ROC-analysis. Sensitivity and specifity of a single variable or the combination of 2 variables as well as cutoff-values for the diagnosis of IDA were calculated. Group comparisons for IDAPatients without IDA were assigned to a control group. Results: Enrolled were 314 patients from an interdisciplinary ICU. RET-He (Area under curve (AUC) 0.846) and Delta-He (AUC 0.807) did indicate IDA more specific () and sensitive () in comparison to serum ferritin (AUC 0.678) and TSAT (AUC 0.754). The detection of functional iron deficiency (FID) occurred in nnn % of cases without anemia. Conclusions: Determination of RET-He and Delta-He allows an increased precision and sensitivity of IDA in the ICU.  

 We adopted the suggestions

Introduction section:

The section is lengthy although the diagnostic problem is well described. It needs to be condensed to its half.

  • Line 32: Avoid unscientific wording such as “Even today”. It is not obvious what do you mean by this or to which condition this expression is directed. It is ok to start the Sentence with “Anemia…”
  • Line 99-102: This statement is your hypothesis, right? So name it: We hypothesize that
  • Line 103-106: Then erase the phrase “To facilitate….”, continue “This study compares …to…

 All implemented as suggested, including shortening the introduction. It now ends in line 80, where it previously went to line 106.

Materials and Methods:

  • This section should be reorganized. Details on the FACS methodology should be described here. Furthermore, the paragraphs and titles need to be clear and comprehensive. For example, you are giving the group allocation within the statistical analysis paragraph, you are mentioning a control group in the classification of anemia.  I suggest to arrange paragraphs for study design and study aim, definition of groups and allocation, lab methods, statistical analysis.

Adopted as suggested

  • Line 149- either you define “diagnostic performance”, or you be more precise especially for the studies endpoint. The reader wants to know exactly what the primary endpoint of your study is.

Adjusted

  • Line 126-131: You devide the subjects of your cohort in diagnostic groups- by which criteria? What is the background to differentiate between ID with out anemia and FID? The group allocation in lines 126-131 came without a reference. Are you trying to tell nutritional from inflammatory based ID? What is the intention behind this separation? It is possible to tell by a combination of Ferritin and TSAT from regular ID without anemia. Explain to the reader why this FID is important to identify and to differentiate from the regular ID without anemia.

We added the criteria in a new table (Table 1) and provided an explanation in the introduction.

  • Lines 157ff : By which statistical procedure the group comparison was done? Multiple group comparison by Kruskall-Wallis should be mentioned here already.

Adjusted

  • Line 173-4- erase the not correct formatted citation in paracentesis. Zweig and Campell are citation 23. Explain in brief how this method works.

 Adjusted

Results:

This paragraph is in general too long. You don’t have to mention the statistical method before you present the results (i.E. the first sentence of 3.2. “Spearman….).

  • Avoid redundancy. All interpretation and data presented as graphs should be erased.
  • The results have to be presented objectively and neutral without explanation or interpretation “ because of their function as acute phase proteins”)- you can comment in the discussion section.
  • Avoid the word “significant”. You defined a level of significance before and that is it.

Rewritten as requested 

  • First paragraph: Again, avoid redundancy: Erase length text passages with a lot of numbers that better are given in a table. The underlying illness such as septic or cardiogenic shock or the leading ICD of your heterogenic collective should be mentioned.

Revised and Table 2 added.

  • Table 1 appears to be all mixed up since alignment and centration errors affect the readability.
  • Use table 1 for the disposition of demographic data, add BMI, Apache-Score or your score for illness severity in Germany, length of ICU stay at enrolment, for example- everything with a connection to the development of anemia.

Now included in Table 2

  • Use a second table for lab parameters only.

Added as Table 3

  • Rename the paragraphs from 3.3 to 3.5 (the use of paragraph titles is not common) by their clinical meaning instead by the method. 3.3. deals with sensitivity and specificity of Ret-He and Delta-He. Discard all text that duplicates the information of the ROC-curve graphs- for example lines 232-239. Erase the part “…, even with simultaneously ongoing inflammatory reactions” in lines 240-241, since you don’t show correlations to inflammation variables in those graphs. There are double dots in the legends graph at the end line 245.

Adjusted as requested.

  • Figure 1 Explain that the term criterion

Done

  • In 3.4 you compare the sensitivity in relation to the specificity of the various variables for the differential diagnosis of IDA and FID. Please shorten the text from line 247 to 268 and erase non-neutral comments. Be describing and explaining instead of advertising. Shift comments such as “For methodical reasons, we must assume that the diagnostic power of serum ferritin is overestimated in this study design, because of the use of this parameter in stratification for the group assignment.“ (lines 253-258) to the discussion. Include the data information in the graphs and legend of Fig 2.

Text adjusted as requested, data information added as Table 4

  • In 3.5 you try to demonstrate the superiority of Ret-He and Delta-He for the differentiation of various combinations of anemia and (functional) iron deficiency and their independency of inflammation. For the reader, this paragraph needs more simplification, less data included in the text and concentation, less interpretation. Shift everything

Adjusted as requested.

  • Legend Fig 3. Reword in something like this: Ability of Ret-He , Delta-He, ferritin and CRP to identify anemic and iron deficient states. Group comparison by Kruskall-Wallis. The colored band marks the reference range.

 Adjusted as requested.

Discussion:

  • In general, you want to say that in comparison of Ferritin and TSAT,  RET-Hand Delta-Hewere able to identify iron deficiency and IDA independent from inflammation. Please rearrange the whole discussion, erase the paragraph titles. First give a short summary of your findings, then discuss why this is important, what is congruent with your findings, is there literature not supporting your results. Then discuss in detail the potential methodological flaws of your method- is it ok to select of group allocation based on parameters that are giving a correlation with the new parameters?

This section has to be convincing and a pleasure to read!

Section has been restructured and rephrased.

Reviewer 2 Report

The authors compared serological parameters (serum ferritin and transferrin saturation (TSAT)) and reticulocyte biomarkers (hemoglobin content (RET-He) and Delta-hemoglobin equivalent (Delta-He)) using 314 patients from the ICU as part of an observational study. They reported that RET-He (Area under curve (AUC) 0.846) and Delta-He (AUC 0.807) showed better diagnostic capabilities than serum ferritin (AUC 0.678) and TSAT (AUC 0.754), especially for the detection of functional iron deficiency (FID), as well as comparable suitability for diagnosing iron deficiency anemia (IDA). The article is generally well-written, and the findings are essential to clinical health. However, some points need to be clarified before publication.

MAJOR Comments

Abstract provided RET-He (AUC 0.846), but AUC was 0.847 in the main text. Please provide consistent results for RET-He and Delta-He.

Please provide the results of some previous research in this field and compare your results with theirs.

It would be better to organize Table 1 for better reading.

Please consider using one standard deviation (SD) for result comparisons.

MINOR Comments

Figure 1 caption has two periods.

The “lowest” should be “the lowest”.

The “a HAA” should be “an HAA”.

The “prooves” should be “proves”.

The “Summery” should be “Summary”.

Author Response

Reviewer 2:

We thank the reviewer for their valuable suggestions how to improve the article. Regarding the individual points, we have addressed them as follows.

MAJOR Comments

Abstract provided RET-He (AUC 0.846), but AUC was 0.847 in the main text. Please provide consistent results for RET-He and Delta-He.

Done

Please provide the results of some previous research in this field and compare your results with theirs.

It would be better to organize Table 1 for better reading.

Done

Please consider using one standard deviation (SD) for result comparisons.

We are not quite sure which results the reviewer refers to here. We provided both standard deviations and IQR for the laboratory parameters to illustrate the distribution of the parameters (for our study, they were found to be distributed in a non-normal fashion). For comparison of AUC, parametric approaches are not customary.

MINOR Comments

Figure 1 caption has two periods.

Adjusted

The “lowest” should be “the lowest”.

Moot through other adjustments

The “a HAA” should be “an HAA”.

Adjusted

The “prooves” should be “proves”.

Adjusted

The “Summery” should be “Summary”.

Moot through other adjustments

Round 2

Reviewer 1 Report

The revised manuscript entitled “Improved diagnosis of iron deficiency anemia in the critically ill via fluorescence flowcytometric hemoglobin biomarkers” reports the sensitivity and specifity of alternate diagnostic for iron deficiency anemia.

The major part of revisions are sufficiently done, the discussion section still needs considerable improvement and overhaul.

Minor criticism

Introduction section:

Line 46 – erase “often”

Line 48 – erythrocytes, and an

Line 55 – erase “or newer”: …in such cases [5,8]. Serological parameters…

Line 57 – “…log ferritin), or hepcidin…”

Line 58 – erase “, but still“: …both need…”

Line 59 – move “and”: “designs, and better laboratory prevalence. Additionally, they are cost consuming.”

Line 62: Erase “ and its recent developments”

LINE 62 /63: ERASE COMMAS

Line 66: Start a new sentence after “(RBC.He).

Line 69: [15], and

Line 78: Erase “, and adjust therapy accordingly”

Line 79: “…reticulocytes parameters with diagnostic standards serum ferritin and TSAT for the diagnosis…”

Materials and Methods

Line 89/90- Prospectively, following informed consent, in-hospital patients…disciplines were enrolled.

Line 106- erase ,which and start a new sentence: “…[20]. The latter are…”

Results

Line 182- please clarify: do you mean : “At study start, 185 () subjects treated on ICU already had been anemic by WHO-definitions. From other enrolled 132 subjects, 30 () acquired anemia in the ICU or in the hospital (HAA)” Overall,  212 () subjects have been discharged from ICU with IDA, FID,or anemia of other cause (25 (11.8%), 60 (28.3%), or 127 (59.9%), respectively)?

You could mention that in the group of anemia, a higher APACHE II and more deaths were registered.  

Line 211- grammatical error- were compared. Erase “Here,”

Line 214- erase “show significant….MCV” after “did not.”

Line 215- correlations

Line 216- erase the interpretation: “, because of their function…”

Line 225- erase the interpretation (whole sentence)

Line 232- for the diagnosis of IDA

Discussion

Unchanged to the first version, the discussion still is lengthy and contains information unneeded for the readers comprehension of this work. The section still could be condensed considerably. On the other side, the critical scientific content needs to be deepend.

The resolved dilemma for anemia diagnosis in the coexistence of inflammation by reticulocyte hemoglobin and its relation to the regular hemoglobin content has appreciated side benefits such as economic and methodological ease.  However, the paragraphs from line 305 to 323 are not needed. The effect of this new method upon the reduction of iatrogenic anemia during the ICU stay can be mentioned as a PBM element with one citation of the PBM concept. However, the statement that ferritin and CRP parameters could be dismissed needs discussion without blowing up the paragraph from Line 324 to 334 further.

There are more questions to discuss that remain unmentioned.  

·      The rate of anemia of other causes seems to be high. Is that in congruence to other studies?

·      The studies characteristics needs to be explained a bit better: The ICU stay was relatively short and the Appache II of the cohort low. What would be the impact of higher scores and other characteristics?

. The methodological question of the first version still is unanswered ( Is it ok to select of group allocation based on parameters that are giving a correlation with the new parameters?)

. The use of an optimality criterion relates the cost of true and false decisions. Could you explain this method and its limitations for the replacement of current standards in iron deficiency anemia standards?

Conclusion

You did not do a cost efficiency calculation nor did you do a time assessment– so you cannot state that the method is “cost‐effective and timesaving” . It is enough that the method enables the detection of iron deficient anemia and subtypes.

Author Response

Review Round 2:

Open Review

English language and style

( ) English very difficult to understand/incomprehensible
(x) Extensive editing of English language and style required
( ) Moderate English changes required
( ) English language and style are fine/minor spell check required
( ) I don't feel qualified to judge about the English language and style

Yes

Can be improved

Must be improved

Not applicable

Does the introduction provide sufficient background and include all relevant references?

(x)

( )

( )

( )

Are all the cited references relevant to the research?

( )

(x)

( )

( )

Is the research design appropriate?

(x)

( )

( )

( )

Are the methods adequately described?

(x)

( )

( )

( )

Are the results clearly presented?

(x)

( )

( )

( )

Are the conclusions supported by the results?

(x)

( )

( )

( )

Comments and Suggestions for Authors

The revised manuscript entitled “Improved diagnosis of iron deficiency anemia in the critically ill via fluorescence flowcytometric hemoglobin biomarkers” reports the sensitivity and specifity of alternate diagnostic for iron deficiency anemia.

The major part of revisions are sufficiently done, the discussion section still needs considerable improvement and overhaul.

Minor criticism

Introduction section:

  • Line 46 – erase “often”
  • Line 48 – erythrocytes, and an
  • Line 55 – erase “or newer”: …in such cases [5,8]. Serological parameters…
  • Line 57 – “…log ferritin), or hepcidin…”
  • Line 58 – erase “, but still“: …both need…”
  • Line 59 – move “and”: “designs, and better laboratory prevalence. Additionally, they are cost consuming.”
  • Line 62: Erase “ and its recent developments”
  • LINE 62 /63: ERASE COMMAS
  • Line 66: Start a new sentence after “(RBC.He).
  • Line 69: [15], and
  • Line 78: Erase “, and adjust therapy accordingly”
  • Line 79: “…reticulocytes parameters with diagnostic standards serum ferritin and TSAT for the diagnosis…”

Materials and Methods

  • Line 89/90- Prospectively, following informed consent, in-hospital patients…disciplines were enrolled.
  • Line 106- erase ,which and start a new sentence: “…[20]. The latter are…”

Results

  • Line 182- please clarify: do you mean : “At study start, 185 () subjects treated on ICU already had been anemic by WHO-definitions. From other enrolled 132 subjects, 30 () acquired anemia in the ICU or in the hospital (HAA)” Overall,  212 () subjects have been discharged from ICU with IDA, FID,or anemia of other cause (25 (11.8%), 60 (28.3%), or 127 (59.9%), respectively)?
  • You could mention that in the group of anemia, a higher APACHE II and more deaths were registered.  
  • Line 211- grammatical error- were compared. Erase “Here,”
  • Line 214- erase “show significant….MCV” after “did not.”
  • Line 215- correlations
  • Line 216- erase the interpretation: “, because of their function…”
  • Line 225- erase the interpretation (whole sentence)
  • Line 232- for the diagnosis of IDA

Discussion

  • Unchanged to the first version, the discussion still is lengthy and contains information unneeded for the readers comprehension of this work. The section still could be condensed On the other side, the critical scientific content needs to be deepend.

  • The resolved dilemma for anemia diagnosis in the coexistence of inflammation by reticulocyte hemoglobin and its relation to the regular hemoglobin content has appreciated side benefits such as economic and methodological ease.  However, the paragraphs from line 305 to 323 are not needed. The effect of this new method upon the reduction of iatrogenic anemia during the ICU stay can be mentioned as a PBM element with one citation of the PBM concept.

This seems to be a misunderstanding from the reviewer’s side. The main effect is not a reduction of iatrogenic anemia, but to reduce the use of packed red blood cells through treating iron deficiency anemia with iron substitution rather than PRBCs. We’ve tried to make this clearer.

  • However, the statement that ferritin and CRP parameters could be dismissed needs discussion without blowing up the paragraph from Line 324 to 334 further.

We do not suggest that ferritin and CRP parameters should be dismissed completely at all. Quite the contrary, we suggest that CRP can be used to initially establish the presence of inflammation. Ferritin may have a role, once iron deficiency in hematopoiesis is established, in characterizing which type of iron deficiency it is, as this will have implications as to what kind of iron substitution should be chosen, oral or intravenous. But there are alternatives for that as well, as discussed. In any case, neither necessarily needs to be measured repeatedly. Cf. also  Auerbach M, Staffa SJ, Brugnara C. Using Reticulocyte Hemoglobin Equivalent as a Marker for Iron Deficiency and Responsiveness to Iron Therapy. Mayo Clin Proc. 2021 Jun;96(6):1510-1519. doi: 10.1016/j.mayocp.2020.10.042. Epub 2021 May 2. PMID: 33952394 and Almashjary, M.N.; Barefah, A.S.; Bahashwan, S.; Ashankyty, I.; ElFayoumi, R.; Alzahrani, M.; Assaqaf, D.M.; Aljabri, R.S.; Aljohani, A.Y.; Muslim, R.; Baawad, S.A.; Bawazir, W.M.; Alharthy, S.A. Reticulocyte Hemoglobin-Equivalent Potentially Detects, Diagnoses and Discriminates between Stages of Iron Deficiency with High Sensitivity and Specificity. J. Clin. Med. 2022, 11, 5675. https://doi.org/10.3390/jcm11195675

  • There are more questions to discuss that remain unmentioned.  
    • The rate of anemia of other causes seems to be high. Is that in congruence to other studies?

In general, the rate of anemia of all causes is very high in ICU settings, reaching up to 98% in some studies (Thomas J., Jensen L., Nahirniak S., Gibney R.T. Anemia and blood transfusion practices in the critically ill: A prospective cohort review. Heart Lung. 2010;39:217–225. doi: 10.1016/j.hrtlng.2009.07.002.)

A distinction between causes requires a differential diagnosis that is not always conducted. The prevalence of various forms of anemia will invariably also vary depending on the general composition of the patient contingent. A general ICU will have different patients as that of more specialized ICUs. The population studied was a general ICU, with surgical and non-surgical patients. It was also an ICU at a non-maximum care hospital, also limiting what kind of patients are to be expected. Literature has FID at 13-35% in general ICU (Muñoz M, Romero A, Morales M, Campos A, García-Erce JA, Ramírez G. Iron metabolism, inflammation and anemia in critically ill patients. A cross-sectional study. Nutr Hosp. 2005 Mar-Apr;20(2):115-20. PMID: 15813395; Patteril MV, Davey-Quinn AP, Gedney JA, Murdoch SD, Bellamy MC. Functional iron deficiency, infection and systemic inflammatory response syndrome in critical illness. Anaesth Intensive Care. 2001 Oct;29(5):473-8. doi: 10.1177/0310057X0102900504. PMID: 11669426) but a large number of COVID-19 patients can easily drive such numbers up (Bellmann-Weiler R, Lanser L, Barket R, Rangger L, Schapfl A, Schaber M, Fritsche G, Wöll E, Weiss G. Prevalence and Predictive Value of Anemia and Dysregulated Iron Homeostasis in Patients with COVID-19 Infection. J Clin Med. 2020 Jul 29;9(8):2429. doi: 10.3390/jcm9082429. PMID: 32751400; PMCID: PMC7464087.)

Conversely, an ICU cohort consisting only of COVID-patients will see a significantly higher rate of FID (Bellmann-Weiler R, Lanser L, Barket R, Rangger L, Schapfl A, Schaber M, Fritsche G, Wöll E, Weiss G. Prevalence and Predictive Value of Anemia and Dysregulated Iron Homeostasis in Patients with COVID-19 Infection. J Clin Med. 2020 Jul 29;9(8):2429. doi: 10.3390/jcm9082429. PMID: 32751400; PMCID: PMC7464087.)

As such, we do not see a specific need to discuss these expectable results in the article.

    • The studies characteristics needs to be explained a bit better: The ICU stay was relatively short and the Appache II of the cohort low. What would be the impact of higher scores and other characteristics?

The median ICU stay way in line with critical care databases (Johnson, A. E. W., Pollard, T. J., Shen, L., Lehman, L. H., Feng, M., Ghassemi, M., Moody, B., Szolovits, P., Celi, L. A., & Mark, R. G. (2016). MIMIC-III, a freely accessible critical care database. Scientific Data, 3, 160035.)

As described above, the hospital is a non-maximum care regional hospital. Patients with more complicated diseases requiring extensive specialty care will be re-routed to other hospitals. The Apache II is comparable with nonseptic patients in other studies (e.g. Basile-Filho A, Lago AF, Menegueti MG, Nicolini EA, Rodrigues LAB, Nunes RS, Auxiliadora-Martins M, Ferez MA. The use of APACHE II, SOFA, SAPS 3, C-reactive protein/albumin ratio, and lactate to predict mortality of surgical critically ill patients: A retrospective cohort study. Medicine (Baltimore). 2019 Jun;98(26):e16204. doi: 10.1097/MD.0000000000016204. PMID: 31261567; PMCID: PMC6617482 )

  •  

Higher scores and longer stay would only increase the risk of iron deficiency, and thus the necessity for its proper diagnosis (cf. the previously mentioned COVID study). Again, we do not feel the article would be improved by expanding on these results, as they are expectable given the patient cohort.

  • . The methodological question of the first version still is unanswered (Is it ok to select of group allocation based on parameters that are giving a correlation with the new parameters?)

We have, in fact, sought to explain the issue in the text – while it may be statistically problematic, it is an almost invariable consequence when looking at parameters measuring physiologically linked phenomena. The definition of the various forms of anemia is what it is, and even resorting to bone marrow samples would not have changed the fact that the iron status in the bone marrow will also to some degree correlate with the iron used in hematopoiesis, and thus the iron in reticulocytes.

The issue is illustrated by a multitude of other studies operating in the same fashion, such as

  • Neef V, Schmitt E, Bader P, Zierfuß F, Hintereder G, Steinbicker AU, Zacharowski K, Piekarski F. The Reticulocyte Hemoglobin Equivalent as a Screening Marker for Iron Deficiency and Iron Deficiency Anemia in Children. J Clin Med. 2021 Aug 9;10(16):3506. doi: 10.3390/jcm10163506. PMID: 34441801; PMCID: PMC8397001.
  • Almashjary, M.N.; Barefah, A.S.; Bahashwan, S.; Ashankyty, I.; ElFayoumi, R.; Alzahrani, M.; Assaqaf, D.M.; Aljabri, R.S.; Aljohani, A.Y.; Muslim, R.; Baawad, S.A.; Bawazir, W.M.; Alharthy, S.A. Reticulocyte Hemoglobin-Equivalent Potentially Detects, Diagnoses and Discriminates between Stages of Iron Deficiency with High Sensitivity and Specificity. J. Clin. Med. 2022, 11, 5675. https://doi.org/10.3390/jcm11195675
  • Auerbach M, Staffa SJ, Brugnara C. Using Reticulocyte Hemoglobin Equivalent as a Marker for Iron Deficiency and Responsiveness to Iron Therapy. Mayo Clin Proc. 2021 Jun;96(6):1510-1519. doi: 10.1016/j.mayocp.2020.10.042. Epub 2021 May 2. PMID: 33952394.

Previous investigations already confirmed the potential of reticulocyte hemoglobin to detect ID verified by bone marrow aspiration staining

  • Rehu M, Ahonen S, Punnonen K. The diagnostic accuracy of the percentage of hypochromic red blood cells (%HYPOm) and cellular hemoglobin in reticulocytes (CHr) in differentiating iron deficiency anemia and anemia of chronic diseases. Clin Chim Acta. 2011 Sep 18;412(19-20):1809-13. doi: 10.1016/j.cca.2011.06.004. Epub 2011 Jun 14. PMID: 21689644.
  • Mehta S, Goyal LK, Kaushik D, Gulati S, Sharma N, Harshvardhan L, Gupta N. Reticulocyte Hemoglobin vis-a-vis Serum Ferritin as a Marker of Bone Marrow Iron Store in Iron Deficiency Anemia. J Assoc Physicians India. 2016 Nov;64(11):38-42. PMID: 27805332.
  • . The use of an optimality criterion relates the cost of true and false decisions. Could you explain this method and its limitations for the replacement of current standards in iron deficiency anemia standards?

 We had expanded on the description of the method in the statistical section already for the previous revision and have added some more detail now. We also had pointed out that the results of the method will vary depending on the composition of the patient population (cf. above discussion of the varying prevalence of iron deficiency). The pertinent reference for the method as such is linked.

Conclusion

  • You did not do a cost efficiency calculation nor did you do a time assessment– so you cannot state that the method is “cost‐effective and timesaving” . It is enough that the method enables the detection of iron deficient anemia and subtypes.

We have removed the pertinent statement. It was based on the fact that the CBC with reticulocytes is done in less than two minutes and the additional cost is less than €1, whereas parameters such as ferritin require, including centrifugation steps, over 30 minutes total measuring time and are reimbursed with more than €10. That being said, we acknowledge we did not do an assessment of the eventual total time and costs.

Round 3

Reviewer 1 Report

Thanks for the new revision and making things clearer to the reader.

The manuscript is acceptable for publication now. your work is important since it demonstrates next to the main topic the diagnosis of iron anemia and iron deficiency in the critical ill that anemia is more frequent than in the recognition of most scientist since underdiagnosed with respext to its underlying cause.

Therefore, I suggest you introduce the point of criticism into the discussion as you did answer:

Probably like this:

The rate of anemia appears to be high in our study. However, anemia of all causes is very high in ICU settings, climbing to 98% in some studies with a longer ICU stay (Thomas J., Jensen L., Nahirniak S., Gibney R.T. Anemia and blood transfusion practices in the critically ill: A prospective cohort review. Heart Lung. 2010;39:217–225. doi: 10.1016/j.hrtlng.2009.07.002.) Frequently, the diagnosis of ICU-associated anemia nor identifies the underlying defiviency nor differentiates iron anemic states from iron deficiency (FID). FID is even higher in special ICU collectives such as COVID-19(Bellmann-Weiler R, Lanser L, Barket R, Rangger L, Schapfl A, Schaber M, Fritsche G, Wöll E, Weiss G. Prevalence and Predictive Value of Anemia and Dysregulated Iron Homeostasis in Patients with COVID-19 Infection. J Clin Med. 2020 Jul 29;9(8):2429. doi: 10.3390/jcm9082429. PMID: 32751400; PMCID: PMC7464087.)

Author Response

Dear Reviewer,

we thank the reviewer for his comments and suggestions. We added your paragraph to Discussion and included the suggested references.

Dur to you comments the manuscript greatly improved.

Thank you so much. Greetings from Germany

Sincerely

Prof. Dr. Christian Hönemann